

# Finite domains cause bias in measured and modeled distributions of cloud sizes

Thomas D. DeWitt[1] and Timothy J. Garrett[1]

[1]Department of Atmospheric Sciences, University of Utah, 135 S 1460 E Rm 819, Salt Lake City, UT 84112, USA;

**Correspondence:** tim.garrett@utah.edu

**Abstract.** A significant uncertainty in assessments of the role of clouds in climate is characterization of the full distribution of their sizes. Order-of-magnitude disagreements exist among observations of such key distribution parameters as the power law exponent and the range over which a power law applies. A study by Savre and Craig (2023) proposed this discrepancy owes in large part to inaccurate fitting methods. Rather than linear regression to a logarithmically-transformed histogram of cloud sizes, an alternative method termed Maximum Likelihood Estimation was recommended. Here, we counter that Maximum Likelihood Estimation is ill-suited to measurements of physical objects like clouds, and that the accuracy of linear regression can be improved with the simple remedy that bins containing less than $\sim 24$ counts be omitted from the regression. Further, we argue that the unavoidably finite nature of measurement domains is a much more significant source of error than has previously been appreciated. Finite domain effects are sufficient to account for previously observed discrepancies among reported cloud size distributions. We provide a simple procedure to identify and correct finite domain effects that could be applied to any measurement of a geometric size distribution of objects, whether physical, ecological, social or mathematical.

## 1 Introduction

The broad range of cloud sizes in the atmosphere poses a significant challenge to the modeling of weather and climate. Small clouds tend to be most numerous while large clouds have more significant meteorological and climate impacts. An approximate balance means that all size classes contribute to overall cloud cover (Wood and Field, 2011), total rainfall (Peters et al., 2009), and the dissipation of buoyant potential energy (Garrett et al., 2018). The commonly used "divide and conquer" approach to the problem isolates a particular spatial scale for study, such as mesoscale convective systems larger than $\sim 100\,\mathrm{km}$ (Houze Jr., 2004), shallow clouds in the trades between 20 to $200\,\mathrm{km}$ (Stevens et al., 2020; Bony et al., 2020), or sub-kilometer cumulus (Koren et al., 2008; Mieslinger et al., 2019). While this approach has practical benefits, it cannot easily be used to address how clouds of all scales interact.

Revealingly, independent of spatial scale or cloud type considered, the measured horizontal dimensions of clouds tend to follow power law distributions such that the number of clouds is proportional to their size to some power (Cahalan and Joseph, 1989; Wood and Field, 2011; Mieslinger et al., 2019; Savre and Craig, 2023). Quantities that follow power law distributions are often described as being "scale-free" or "scale-invariant", meaning that there is no "characteristic" object scale as there would be defining, for example, an exponential or Gaussian distribution. Power law behavior is in fact quite general among





physical, social, and biological systems, applying to e.g. meteor diameters, neuronal firing, personal incomes, city populations, and forest sizes (Buzsáki and Draguhn, 2004; Newman, 2005; Bettencourt et al., 2007; Saravia et al., 2018).

Quantities that exhibit scale-free behaviors, however deterministically complicated they may be, allow for an important mathematical simplification. Namely, phenomena measured at any one scale shed light on the behavior at others. They also present a practical challenge, which is the unavoidable limitation that geometrically defined objects must inevitably be measured within a domain of some finite size; in other words, a domain that is *not* scale-free. The domain enforces a maximum scale for object measurement – the size of the domain – and this may not reflect the maximum scale that the objects can attain.

For cloud areas $a$, we recently argued that improper consideration of the scale of the measurement domain has contributed to wide discrepancies in the reported nature of cloud area distributions, and in particular to the upper bound $a_{\max}$ to which a power law can be claimed to apply (DeWitt et al., 2024). Because clouds cannot be arbitrarily small, there must also be a lower bound $a_{\min}$ to the power law regime, one that has not yet been determined but could approach the Kolmogorov microscale of $\sim 1\,\mathrm{mm}$, below which turbulent circulations are damped by viscous forces. Between $a_{\min}$ and $a_{\max}$, the power law regime can be represented by the probability distribution

$$n(a) \propto a^{-(\alpha+1)}; \qquad a_{\min} < a < a_{\max}, \tag{1}$$

where $\alpha$ is a constant, indicating $n(a)$ is linear on doubly-logarithmic axes. The upper bound at $a_{\max}$ represents a "scale break" beyond which studies generally find the distribution is "cut off" by a regime following either a steeper power law with a larger value of $\alpha$ or an exponential (Cahalan and Joseph, 1989; Benner and Curry, 1998; Neggers et al., 2003; Peters et al., 2009; Mieslinger et al., 2019; van Laar et al., 2019; Christensen and Driver, 2021; Savre and Craig, 2023).

Estimates of the location of the scale break at $a_{\max}$ differ widely. This uncertainty has underappreciated implications for studies of the role of clouds in climate because the integral $\int_{a_{\min}}^{a_{\max}} a n(a) da$, which is the total cloud amount, is sensitive to the scale break location. For example, a cutoff regime at areas of order $\sim 10\,\mathrm{km}^2$, as suggested by some (Cahalan and Joseph, 1989; Benner and Curry, 1998; Neggers et al., 2003; Savre and Craig, 2023), would imply clouds larger than $\sim 4000\,\mathrm{km}^2$ would be so rare that they would contribute negligibly to the total, while other findings suggest that such large clouds contribute approximately 50% to the global cloud cover (Wood and Field, 2011; DeWitt et al., 2024).

The power law exponent $\alpha$ for cloud areas is also highly uncertain, with similar implications for the relative role of different cloud types. The exponent determines the relative numbers of small and large clouds. Values close to unity (e.g. Peters et al., 2009; Wood and Field, 2011; Mieslinger et al., 2019; DeWitt et al., 2024) imply clouds of all orders of magnitude contribute equally to the total cloud cover, in which case small clouds that are often left unresolved by models and measurements may be an important omission. Conversely, values less than unity (e.g. Cahalan and Joseph, 1989; Benner and Curry, 1998; Neggers et al., 2003; Koren et al., 2008; Yamaguchi and Feingold, 2013; Bley et al., 2017; Senf et al., 2018; van Laar et al., 2019; Savre and Craig, 2023) indicate large clouds dominate the total area and so remain a reasonable subject for more focused study.

The lack of consensus among studies on the value of $\alpha$ may owe to differences in the dominant cloud type that was considered, or to how diurnal variability affects $a_{\max}$ (van Laar et al., 2019). But even if temporal and spatial variability of the size distribution exists, there remains a necessary prerequisite to measuring such variability, which is to first ensure the size distri-



bution is being accurately measured in the first place. To this end, Savre and Craig (2023) recently argued that the lack of con-
sensus among prior measurements of cloud sizes owes to the use of inaccurate statistical methods to fit power law distributions.
In particular, they showed that the common method of fitting a least-squares linear regression to a logarithmically-transformed
histogram of cloud areas can lead to biased measurements of $\alpha$.

     Here, we argue that the alternative methodology recommended by Savre and Craig (2023) is not strictly appropriate for

cloud measurements. Regardless, we show there is a more important issue, namely whether past studies properly accounted
for the finite size of the study domain. A finite domain size is a general problem for measuring scale-free quantities. Recently,
Serafino et al. (2021) argued that scaling properties of networks can be obscured by such finite-size effects, causing a truly
scale-free network to appear non-scaling.

     Similarly, cloud sizes must necessarily be measured within a non-scaling finite domain. It is easy to appreciate that the area

of clouds larger than the domain size cannot be measured. A more subtle effect is that the measured numbers of clouds of
a given area, even those smaller than the domain area, is highly sensitive to whether clouds that cross the domain edge are
included or removed in the measured distribution (an example is shown in Fig. 1). We term such clouds "truncated clouds" as
they appear effectively truncated by the domain edge, with only the portion of the cloud lying within the domain available to
be measured.

Whether truncated clouds are included or removed from distribution fits is an issue rarely mentioned in past studies, but
those that do consider the effect tend to remove truncated clouds without applying any correction factor (e.g. Peters et al.,
2009; Christensen and Driver, 2021). One exception is a study of one-dimensional cloud chords by Wood and Field (2011),
who found that the removal of clouds truncated by the domain edge leads to an undercounting of large clouds relative to what
would be measured in a larger domain. For cloud areas, it may be hypothesized that a similar effect could explain the observed

differences in measurements of $a_{\mathrm{max}}$.

     In this study, Sect. 2 reconsiders the hypothesis proposed by Savre and Craig (2023) that discrepancies in distribution
parameters can be largely explained by improper methods used to fit a power law distribution to measurements of cloud
sizes. Sect. 3 then examines how the choice of either including or removing clouds truncated by the domain edge can change
the measured cloud size distributions. We suggest such methodology may bias measured distribution parameters and offer

recommendations for future studies that measure any object size distribution within a finite domain, both for clouds or any
other geometrically-defined objects.

## 2   Fitting power law distributions to empirical data

The most straightforward method to fit a power law to empirical measurements of cloud areas is to bin the data into discrete
bins of constant width $\delta a$, resulting in a discrete set of counts $n_i$ for each bin $i$. The logarithm of Eqn. 1 is the linear equation

$\log n_i = -(\alpha + 1)\log a + const.$, so a line can be fit to $\log n_i$ vs. $\log a$ to estimate $\alpha$ using a least-squares linear regression.

     Goldstein et al. (2004) showed that such a linear regression-based estimate can be biased by up to 36% relative to the known
value in computer-generated power law distributed data. "Logarithmic binning", with bins of exponentially increasing width



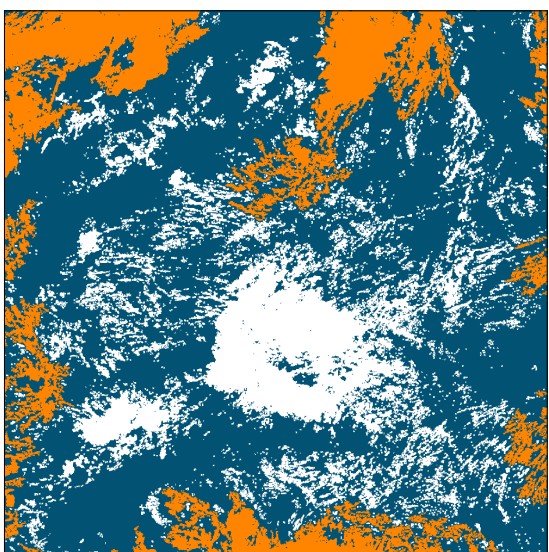

**Figure 1.** An example cloud mask derived from GOES satellite imagery, where cloudy pixels are white or orange and clear pixels are dark blue. Clouds which are truncated by the domain edge are marked in orange. The areas of such "truncated clouds" cannot be properly quantified as some unknown portion lies beyond the measurement domain.

or constant $\delta \log a$, increases bin counts $n_i$ at large $a$. Increasing $n_i$ reduces "statistical error", which is the standard deviation of $n_i$ evaluated over many hypothetical realizations of a given experiment. This reduction in statistical error enables more

accurate estimates of $\alpha$ (White et al., 2008). In the case of logarithmic binning, the calculated slope of a histogram is $-\alpha$ rather than $-(\alpha+1)$ because the number of clouds in a given bin is the bin width, which is proportional to $a$, times the distribution in that region, which is $a^{-(\alpha+1)}$ (mathematically, $dn/d\ln a = a\,dn/da$).

However, even with logarithmic binning, linear regression has been found to produce a biased estimate of $\alpha$ relative to a known value for empirical tests that use computer-generated power law distributed data (Goldstein et al., 2004; White et al.,

2008; Clauset et al., 2009). Nonetheless, linear regression – whether to linear or logarithmically-spaced bins – remains a commonly employed method in cloud studies (e.g. Wood and Field, 2011; Yamaguchi and Feingold, 2013; Bley et al., 2017; Senf et al., 2018).

There are two other linear regression-based approaches worth mentioning, namely cumulative distributions and rank-frequency plots, both of which approximate the integral $\int_{a_{\min}}^{a} n(a')da'$. Fitting a linear regression to such plots has been argued to be su-

perior to fitting a linear regression to a histogram of counts (Clauset et al., 2009). Such approaches work well for unbounded power law distributions with $a_{\max} \to \infty$, but for a truncated power law distribution with finite $a_{\max}$ the cumulative distribution is not linear even with doubly-logarithmic axes (Savre and Craig, 2023). The nonlinearity implies a linear regression would be inappropriate to estimate $\alpha$.



An alternative method of fitting a power law to data, Maximum Likelihood Estimation, is argued on empirical grounds to be generally more accurate than linear regression-based approaches (Goldstein et al., 2004; Newman, 2005; White et al., 2008; Clauset et al., 2009). Maximum Likelihood Estimation employs the "likelihood function", which estimates the probability of observing the measured data given many different possible power law distributions. The distribution that is the best fit is the one that maximizes the likelihood function. Savre and Craig (2023) argued some of the disagreement between prior measurements of cloud size distributions could be resolved through the use of Maximum Likelihood Estimation rather than linear regression-based approaches.

Although Maximum Likelihood Estimation is now commonly used to establish power law behavior, a key requirement is that individual measurements must be statistically independent, a condition often not satisfied in physical systems such as naturally occurring networks (Serafino et al., 2021) and clouds (Garrett et al., 2018). Because cloud formation is constrained by the total available moisture, energy, and space, individual cloud areas are not physically independent and this appears in the statistics. For example, a large but rare cloud that covers over half of a given measurement domain makes it impossible to observe another similarly-sized cloud because a second large cloud could not fit inside of the domain. Thus, the first observation (i.e. the large cloud) alters the probability of the next observation, which violates statistical independence. Similarly, a finite amount of total available energy or moisture make future cloud formation contingent on what has occurred in the past. Maximum Likelihood Estimation-based methods, like those used in Savre and Craig (2023), are therefore inappropriate for measuring cloud size distributions.

Furthermore, evidence supporting the superiority of Maximum Likelihood Estimation put forth by Goldstein et al. (2004), White et al. (2008), and Clauset et al. (2009) was obtained from numerical experiments using synthetic data generated from an unbounded power law (i.e. $a_{\max} \to \infty$ in Eqn. 1). In this case, the likelihood function may be analytically maximized, resulting in a simple formula that can be used to estimate $\alpha$. For a truncated power law with finite $a_{\max}$, however, the likelihood function must instead be numerically maximized (Savre and Craig, 2023), introducing much more complexity and computational expense to the analysis (Hanel et al., 2017), especially when compared to a least-squares linear regression.

In fact, because the truncation at $a_{\max}$ removes the portion of the distribution at large $a$ that contains the most statistical error, it might be argued that linear regression-based approaches are more accurate for power laws that are bounded, as they inevitably are for clouds. Indeed, Goldstein et al. (2004) found that when linear regression was applied to only the smallest five linearly-spaced bins, which effectively truncated the distribution at the upper limit of bin five, the power law exponent was estimated accurately relative to the known value.

To evaluate the accuracy of the linear regression approach for fitting a power law with finite $a_{\max}$, as is relevant for any physical dataset, we randomly sample values for $a$ from a synthetic truncated power law distribution (Eqn. 1) with parameters $\alpha = 1$, $a_{\min} = 10$, and $a_{\max} = 1000$, which are close to what might be measured for cloud sizes. This is accomplished by first drawing $N$ values $a_i$ from an unbounded power law ($a_{\max} \to \infty$) using the Python package `powerlaw` (Alstott et al., 2014) and then removing and re-drawing values larger than $a_{\max}$ until all $N$ values lay within $(a_{\min}, a_{\max})$. This process is repeated until 200 "samples" were created, each with $N = 1000$, $3000$, or $10000$ values. Samples then are binned into 30, 100, or 300 logarithmically-spaced bins and a "minimum bin count" threshold applied, which removes any bin with a count lower than





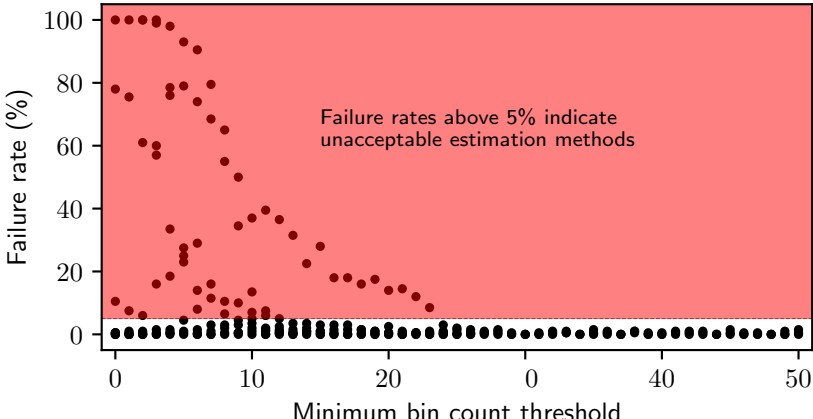

**Figure 2.** Failure rates for fitting $\alpha$ to synthetic data using a linear regression to logarithmically-spaced bins, as a function of the minimum required count in each bin. Each point represents a unique combination of number of bins, sample size, and minimum bin count threshold. Minimum bin count thresholds greater than 24 always ensure accurate estimates of $\alpha$, while smaller thresholds sometimes produce inaccurate estimates. Low failure rates can occur for low minimum bin count thresholds when only a small fraction of bins have low counts. In these cases, most bins are unbiased, and the regression is overwhelmed by the bins with more counts.

a range of specified thresholds between 0 and 50. A fit to each sample is then performed only if the remaining bins span

at least one order of magnitude in $a$. This requirement is necessary for any fitting method because power law distributions fundamentally describe systems spanning many scales (Newman, 2005), but is less stringent than the two orders of magnitude span recommended by Stumpf and Porter (2012).

Estimated values of the power law exponent, denoted $\hat{\alpha}$ to avoid confusion with the specified value $\alpha$, are determined by fitting a least-squares linear regression to the bins satisfying the above criteria. Statistical uncertainty $\varepsilon$ associated with fitted

values $\hat{\alpha}$ is estimated using the Python package `scipy` (Virtanen et al., 2020) as two standard errors on the linear regression, corresponding to a 95% confidence interval. For each combination of sample size, number of bins, and minimum bin threshold, 200 samples are generated. A "failure rate" is calculated as the fraction of estimates that do not include the true value $\alpha = 1$ within their 95% uncertainty range:

$$\text{failure  rate} \equiv \frac{\text{count  of }\; \hat{\alpha} \notin (\alpha - \varepsilon, \alpha + \varepsilon)}{200} \qquad (2)$$

We define an "accurate" estimate method as one whose failure rate is less than 5%. Select tabular results are listed in Appendix D.

As shown in Fig. 2, if the minimum bin count threshold is less than 24, $\alpha$ cannot be accurately estimated using a linear regression technique, in agreement with what was argued by Goldstein et al. (2004), White et al. (2008), and Clauset et al. (2009). However, regardless of sample size or the number of bins, if the regression is only applied to bins with counts of at





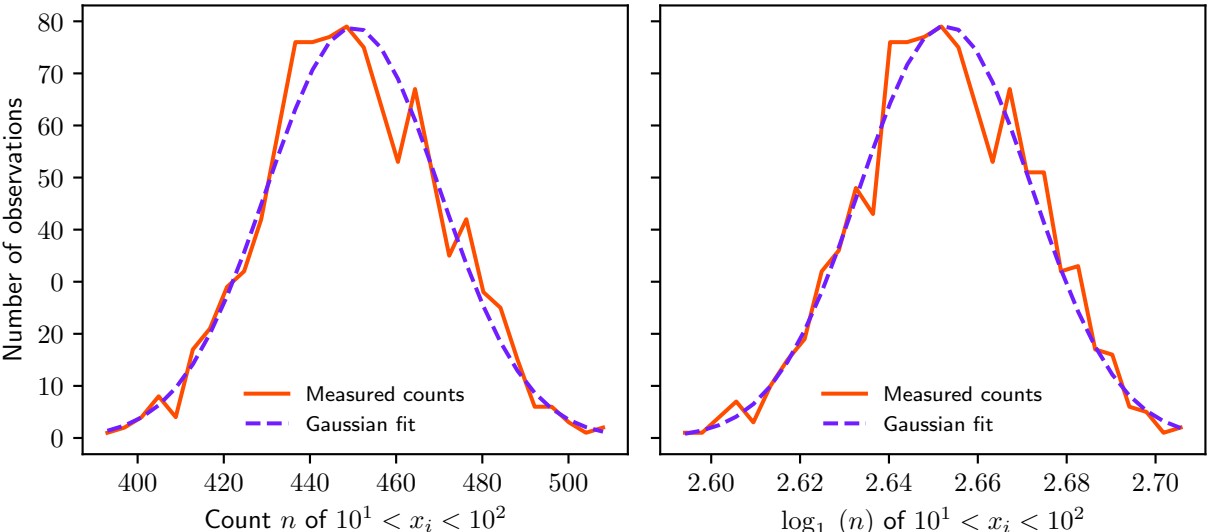

**Figure 3.** Statistical error in measured counts $n_i$ within a bin bounded by 10 and 100 for a collection of 1000 samples, each containing 5000 randomly-generated power law distributed random variables $x_i$ with exponent $\alpha = 1$ (Eqn. 1). Each sample has a count $n_i$ in the bin, and the plot shows a histogram of these counts $n_i$ for all 1000 samples. The plot is thus a "histogram of histograms". The p-value for normality using a Kolmogorov-Smirnov test is 0.333 for $n$ and 0.326 for $\ln n$, indicating the null hypothesis of Gaussian variability cannot be excluded using a 95% confidence threshold in either case. Table A1 shows p-values for more combinations of bin location and sample size.

least 24, estimates of $\hat{\alpha}$ determined from linear regression lie outside of uncertainty bounds less than 5% of the time, which is consistent with a 95% confidence threshold. In this sense, they are accurate.

Applying a simple rule that least-squares linear regression only be applied to those bins with sufficiently large counts may seem obvious: estimating *any* statistical measure using a very small sample tends to result in error. In this particular case of statistically independent measurements, the low failure rates for high minimum bin count thresholds can be understood in

terms of the Central Limit Theorem, where the successive measurement and binning of power law distributed variables can be interpreted as a counting process (see Appendix A). Provided bin counts $n_i$ exceed approximately 24, statistical error in both counts and the logarithm of the counts follows a Gaussian distribution (Fig. 3). In this case, general-purpose linear regression packages that assume Gaussian error at each point may be used to accurately estimate the exponent of a power law distribution.

In summary, whether binning is done linearly or logarithmically, previously calculated values of $\alpha$ for cloud area distributions

that are power law distibuted may be biased, but only if bins with fewer than 24 counts are included in the regression. A very simple fix is to omit such bins. Other studies that estimated $\alpha$ over a range of scales that exclusively included large bin counts (e.g Benner and Curry, 1998; Cahalan and Joseph, 1989; Wood and Field, 2011; DeWitt et al., 2024) may have obtained estimates of $\alpha$ that were reliable. In fact, Mieslinger et al. (2019) estimated $\alpha$ for shallow cumulus using both a linear regression to logarithmically-spaced bins and Maximum Likelihood Estimation, finding both fitting methods produced similar results.





Clauset et al. (2009) state that, while a histogram that is linear when logarithmically transformed is not *sufficient* to identify a power law distribution, linearity is certainly *necessary*. The challenge with cloud sizes is that some portions of the cloud size distribution appear linear in some studies but are clearly nonlinear in others. For example, Cahalan and Joseph (1989); Benner and Curry (1998) and Neggers et al. (2003) all find a scale break at $\sim 1\,\mathrm{km}^2$ where a power law regime transitions to an exponential or a different power law with a much larger value of $\alpha$. Either case indicates a clear nonlinear portion of the

doubly-logarithmic histogram at or beyond the scale break. This is in disagreement with other studies that find linear power law scaling up to $\sim 10\,\mathrm{km}^2$ or $\sim 100\,\mathrm{km}^2$ (van Laar et al., 2019; Savre and Craig, 2023), and especially with the findings of power law scaling extending beyond $10^5\,\mathrm{km}^2$ (Wood and Field, 2011; Christensen and Driver, 2021; DeWitt et al., 2024). Differences in the choice of fitting method used, whether Maximum Likelihood Estimation or regressions to linearly or logarithmically spaced bins, cannot explain these differences in measured $a_{\max}$. Next, we explore instead whether the differences could be

explained by the improper treatment of clouds truncated by the edge of the measurement domain.

## 3   How a finite domain changes measured size distributions

Truncated clouds, which span the domain edge (Fig. 1), present a conundrum. If one wants to accurately measure the size distribution within a finite domain, should one remove them from consideration, risking undercounting clouds in some size classes, or should the clouds be included, risking inaccurate area measurements? To investigate the magnitude of this truncation

effect, we explore measured size distributions for various domain sizes.

### 3.1   Atmospheric cloud measurements, the percolation model, and domain subsampling

For measurements of atmospheric clouds, we use cloud mask data from the Advanced Baseline Imager (ABI) aboard the GOES-West (GOES-17) satellite. GOES-West is a geostationary satellite centered at 137° West with a nadir imaging resolution of approximately $2\,\mathrm{km}$. A pre-processed cloud mask product that attempts to identify every pixel as "cloudy" or "clear" is used,

and so each "image" is a binary array of pixels specified as 1 for cloudy or 0 for clear. A total of 10 processed images are used, each taken at local noon (2100 UTC) between 1 June and 10 June 2021.

    We use the $2000 \times 2000$ pixels located in the center of the image and approximate all pixel dimensions as $2\,\mathrm{km} \times 2\,\mathrm{km}$, which underestimates the true pixel length dimensions by at most 12%. The chosen domain is in the central Pacific between longitudes of 117°W and 157°W and latitudes of 19°S and 19°N. There are no missing data for the domain and time period

considered.

    We also consider size distributions for more idealized objects. The uniform square lattice, adopted from percolation theory, is a two-dimensional square lattice where every site (or cell) is occupied with uniform probability $\mathbb{P}$. "Clusters" are defined as regions of adjacent occupied sites (Stauffer, 1992), and their area $a$ is defined as the number of occupied sites in a single cluster. The mean cluster area $\langle a \rangle$ tends to increase with increasing $\mathbb{P}$ because high site occupation probability increases the

likelihood of site connection (Stauffer, 1992).



A central result of percolation theory is that, as $\mathbb{P}$ approaches a critical point $\mathbb{P}_c \approx 0.592746\ldots$, $\langle a \rangle$ tends to infinity and the distribution of cluster areas follows a power law $n(a) \propto a^{-\tau}$ where $\tau = 187/91$. The power law is only exact in the limit of large clusters and an infinite lattice, but serves as a close approximation for the size distribution of clusters that are larger than about 10 to 20 sites (Stauffer, 1992). In finite lattices, the size of the largest cluster is limited by the size of the lattice, and so

the power law regime cannot extend to arbitrarily large scales as it does for an infinite lattice. This "cut off" is often modeled by an exponential function $n(a) \propto a^{-\tau}e^{-a/a_c}$ where $a_c$ is the characteristic area of the largest clusters, a function of lattice size (Stauffer, 1992).

The percolation model is useful here for studying distributions of object size distributions in finite domains because the distribution of cluster sizes is known exactly. In particular, any deviation from power law scaling at the large end of the cluster

size distribution is known to be because the lattice is of finite size. Models similar to the uniform square lattice used here have also been previously leveraged to explain the fractal dimension of precipitating regions (Peters et al., 2009) and of the power law scaling in cloud sizes itself (Savre and Craig, 2023).

We simulate three $10,000 \times 10,000$ percolation lattices at the percolation threshold $\mathbb{P} = 0.592746$. For both the GOES-derived cloud masks and the percolation lattices, clouds or clusters are defined according to the convention that adjacent pixels

are considered connected and diagonals are not. This is standard practice in both percolation theory (Stauffer, 1992) and in past cloud studies (e.g. Kuo et al., 1993; Wood and Field, 2011). Individual object areas are calculated by summing connected pixel areas, and an object is flagged as "truncated" if it is connected to the lattice boundary (Figure 1).

To test how the domain or lattice size affects the measured area distributions, the binary arrays representing cloud fields or percolation lattices are subdivided as follows: if the shape of the original array is $L \times L$ grid points, with $L = 10,000$ for the

percolation lattices and $L = 2000$ for the GOES West images, sub-arrays are created by choosing a value $q$ and dividing the original array into sub-arrays of size $L/q \times L/q$. We use values of $q \in \{20, 100, 200\}$ for the percolation lattices and values of $q \in \{10, 40, 100\}$ for GOES images. Thus the percolation sub-arrays have side lengths of 500, 100, or 50 grid cells, and the GOES sub-arrays have side lengths of 200, 50, or 20 pixels.

## 3.2 Measured size distributions as a function of domain truncation effects

For each subdomain considered in the cloud imagery, if truncated clouds are removed from the size distributions, bin counts are increasingly undercounted at larger object areas as shown in Fig. 4. A spurious scale break is introduced at these sizes that resembles an "exponential tail", a functional form suggested by Savre and Craig (2023) as being a real characteristic of clouds. Locations of the spurious scale breaks, like those proposed in the literature, span several orders of magnitude but depend only on the domain size. A scale break is introduced because larger clouds are more likely to be truncated and therefore to be

removed from the analysis (Figure 6), particularly for small domains. The clouds need not be particularly large to be affected, as the scale break appears at suprisingly small cloud areas occupying just $\sim 1\%$ of the subdomain area.

Alternatively, if truncated clouds are included in the histogram, they are placed in a smaller size bin than that in which they belong. This leads to an *overcount* for all bins, particularly for large clouds, and a spurious local maximum in cloud frequency for clouds with areas close to the domain area.





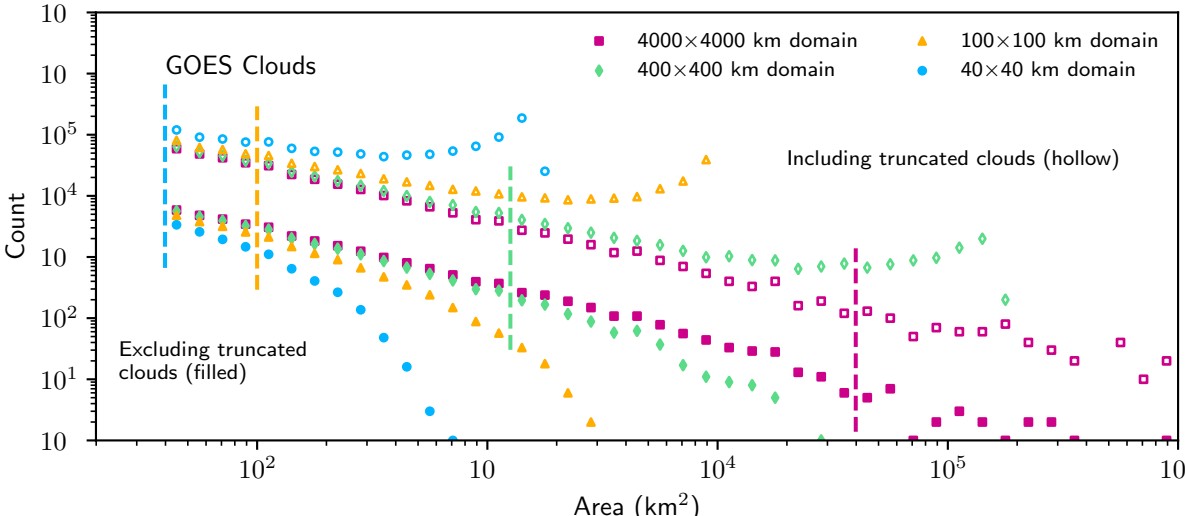

**Figure 4.** Histograms of cloud areas for several sizes of subdomain from GOES West. Filled shapes indicate histograms which do not include truncated clouds, while hollow shapes include truncated clouds. Hollow shapes are offset vertically by a factor of 10 for clarity. The vertical dashed lines mark the first bin in which 50% of objects are truncated by the domain edge for each domain size.

The effect of miscounting large clouds in a finite domain is also mirrored in the percolation lattices, where either a cutoff regime (an undercounting) or a local maximum (an overcounting) is introduced to the size distribution, respectively (Fig. 5). Because percolation clusters are known to follow a power law size distribution, the undercounting or overcounting can only be caused by the finite size of the lattice. This illustrates how truncation effects are not limited to atmospheric clouds but could affect measured size distributions of any phenomenon that is measured within a finite domain.

The simple remedy of calculating $\alpha$ by fitting a power law over a subjectively defined relatively linear region of the distribution, as is often done, can lead to an overestimate of $\alpha$ if truncated clouds are removed, and an underestimate if they are included. As an example, Figure 7 depicts a hypothetical scenario for GOES cloud areas that are measured within a $100 \times 100\,\mathrm{km}$ subdomain. Calculated values of $\alpha$ within this range of scales are significantly affected by whether truncated clouds are included or removed (Table 1). Regardless of whether least-squares linear regression or Maximum Likelihood Estimation is used, including truncated clouds in the fit for $\alpha$ leads to an underestimate of 36% and 19%, respectively, while excluding them leads to an overestimate of 24% and 20%, respectively, relative to values calculated for the full $4000 \times 4000\,\mathrm{km}$ domain. Nonetheless, it is clear from Fig. 7 that both approaches remain well approximated by a power law distribution, and so the truncation effect could easily be missed, leading to reported power law behavior with a value of $\alpha$ that is a significant departure from the true value that would have been measured if the domain had been larger.

We recommend, as a simple solution for the errors introduced by domain truncation effects, to remove from the analysis any bin with a sufficient number of truncated clouds $n_{\mathrm{truncated}}$ relative to the total in that bin $n_{\mathrm{total}}$. Because larger clouds are more



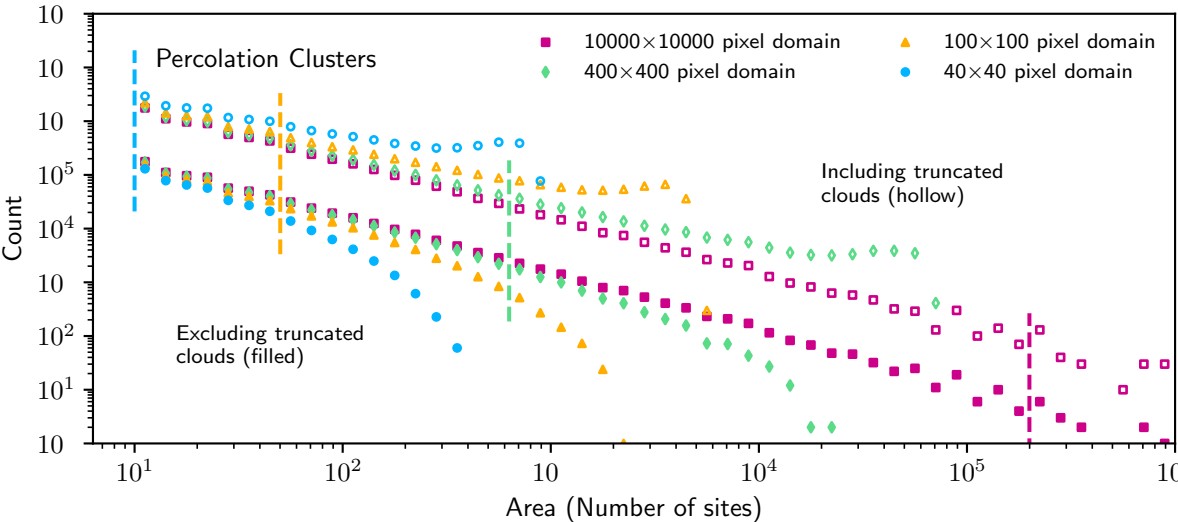

**Figure 5.** As for Figure 4, but for cluster areas in the percolation lattices.

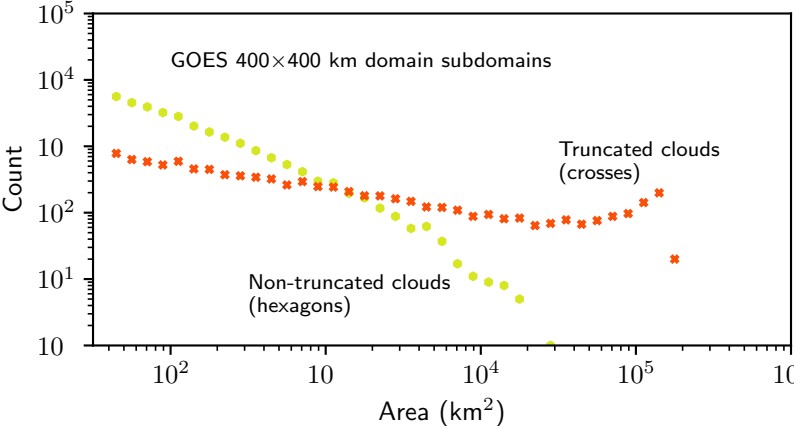

**Figure 6.** Histogram of cloud areas measured in the $400 \times 400$ km subdomains, as in Fig. 4, but separated into those that are truncated by the edge of the domain (crosses) and those that are not (hexagons). At small areas, the number of truncated clouds is negligible compared to the number of non-truncated clouds, but at larger areas the pattern reverses and the number of non-truncated clouds becomes negligible relative to the number of truncated clouds.



**Table 1.** Fits from the hypothetical scenario where cloud areas are measured within the $100 \times 100 \, \text{km}$ GOES subdomains and fits are obtained over a subjectively defined linear region (Fig. 7). Fits are obtained using both linear regression (LR) as described in the text and Maximum Likelihood Estimation (MLE) as described by Savre and Craig (2023). Errors for MLE fits are calculated using a standard bootstrapping procedure and correspond to the 95% confidence interval. For comparison, fits to clouds measured in the full domain are included as "truth". "Differences" is the difference between $\hat{\alpha}$ between the two domain sizes and is expressed in units of standard errors as calculated in the subdomains.

| Domain Size | Fit Range $(a_{\min}, a_{\max})$ | Excluding Truncated LR $\hat{\alpha}$ | Including Truncated LR $\hat{\alpha}$ | Excluding Truncated MLE $\hat{\alpha}$ | Including Truncated MLE $\hat{\alpha}$ |
|---|---|---|---|---|---|
| 4000×4000 km | (20km, 38070km) | $0.93 \pm 0.05$ | $0.90 \pm 0.04$ | $0.97 \pm 0.01$ | $0.96 \pm 0.01$ |
| 100×100 km | (20km, 800km) | $1.2 \pm 0.2$ | $0.7 \pm 0.2$ | $1.22 \pm 0.02$ | $0.79 \pm 0.01$ |
| Difference | | $3.0\sigma$ | $-2.7\sigma$ | $33.1\sigma$ | $-29.2\sigma$ |

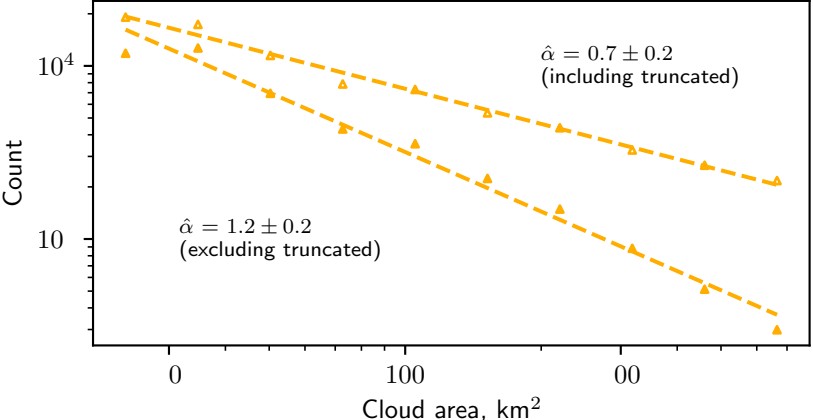

**Figure 7.** Example of how a measurement $\hat{\alpha}$ of the power law exponent could be biased by whether or not truncated clouds are included in the analysis. The histograms shown are a subset of the $100 \times 100 \, \text{km}$ subdomains from GOES shown in Fig. 4. This region is heavily influenced by the choice of including truncated clouds. Fits for $\alpha$ are shown in Table 1.





**Table 2.** Estimated values of $\alpha$ ($\hat{\alpha}$) to cloud areas measured within the full domain and subdomains over the region where $n_{\text{truncated}}/n_{\text{total}} < 0.5$ as a function of the choice of including or excluding truncated clouds in the fit. Only those subdomains in which the fitting region spans at least one order of magnitude are included. Fits are obtained using both linear regression (LR) as described in Sect. 2 and Maximum Likelihood Estimation (MLE) as described by Savre and Craig (2023). Errors for MLE fits are calculated using a standard bootstrapping procedure and correspond to a 95% confidence interval.

| Domain Size | Fit Range $(a_{\min}, a_{\max})$ | Excluding Truncated LR $\hat{\alpha}$ | Including Truncated LR $\hat{\alpha}$ | Excluding Truncated MLE $\hat{\alpha}$ | Including Truncated MLE $\hat{\alpha}$ |
|---|---|---|---|---|---|
| **GOES cloud masks** | | | | | |
| 4000×4000 km | $(80\text{km}^2, 36912\text{km}^2)$ | $0.94 \pm 0.05$ | $0.90 \pm 0.05$ | $0.95 \pm 0.02$ | $0.92 \pm 0.02$ |
| 400×400 km | $(80\text{km}^2, 1481\text{km}^2)$ | $1.0 \pm 0.1$ | $0.8 \pm 0.1$ | $1.02 \pm 0.03$ | $0.85 \pm 0.02$ |
| **Percolation lattices** | | | | | |
| Exact result $187/91 - 1$ | | $1.055$ | $1.055$ | $1.055$ | $1.055$ |
| 10000×10000 site | $(20, 476306)$ | $1.05 \pm 0.03$ | $1.01 \pm 0.04$ | $1.05 \pm 0.01$ | $1.04 \pm 0.01$ |
| 400×400 site | $(20, 637)$ | $1.10 \pm 0.04$ | $0.95 \pm 0.04$ | $1.09 \pm 0.01$ | $0.970 \pm 0.009$ |

likely to be truncated by the domain edge (Figure 6), this procedure effectively removes the large end of the size distribution from the fit. Conveniently, in practice this procedure sometimes also enforces the minimum bin count threshold of 24 that is necessary for reliable linear regression-derived fits for the power law exponent.

In Table 2, estimates of $\alpha$ are listed for the region where $n_{\text{truncated}}/n_{\text{total}} < 0.5$ for a series of of subdomains created from the GOES cloud masks and the percolation lattices. Although imperfect, when a 50% threshold is used, fitted values for $\alpha$ are much less sensitive to the choice of fitting method or to whether truncated clouds are included or removed. Errors in fitted values of $\alpha$ can be further reduced by increasing the range of scales measured, accomplished either by increasing the domain size or decreasing the pixel size.

In principle, because the 50% threshold removes larger objects in the distribution that may be of scientific interest, an algorithm could be devised to correct cloud truncation effects. One such algorithm was used by Wood and Field (2011), however it was assumed that clouds are square shaped. In general, any correction algorithm requires some similarly questionable assumption, and so considerable caution should be taken when devising such an algorithm. This issue is further discussed in Appendix B.

**3.3    Periodic domains**

One commonly employed method for reducing artifacts caused by domain boundaries in cloud modeling simulations is to utilise doubly-periodic simulations that allow fluxes out one side of the numerical grid to re-enter on the opposite (e.g. Neggers et al., 2003; Yamaguchi and Feingold, 2013; Garrett et al., 2018). Unfortunately, even without a domain edge, simulations with periodic domains still suffer from a finite domain area that modifies the cloud size distribution. For example, consider





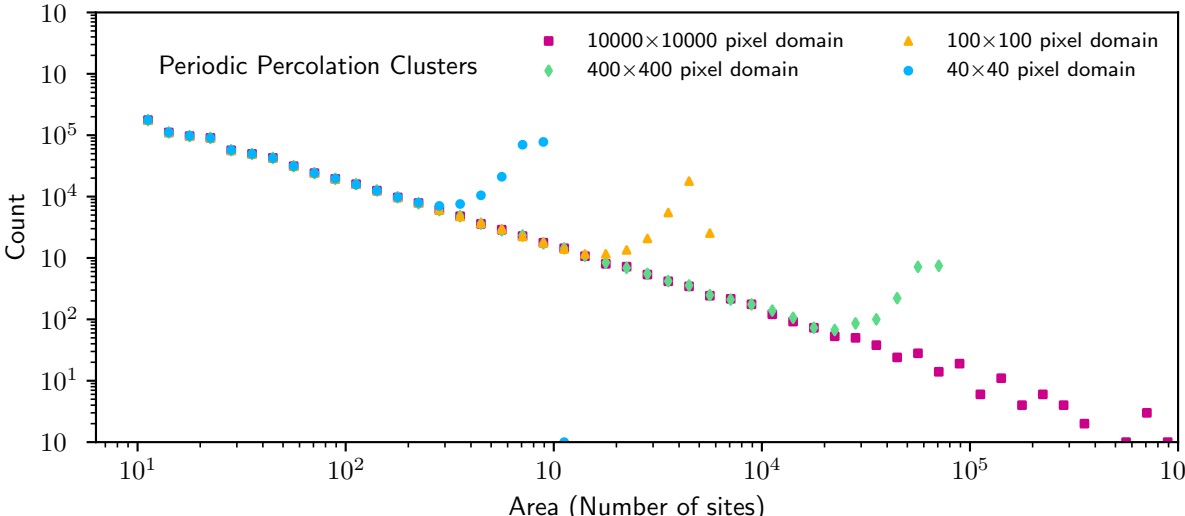

**Figure 8.** Histogram of cluster areas in doubly-periodic percolation lattices for several domain sizes.

the limiting case of a model composed of a single horizontal grid cell. Even with a periodic domain that maintains flux conservation laws, the cloud size distribution would nonetheless be unphysically constrained to one possible cloud size, leaving $\alpha$ undetermined.

The impact of employing periodic domains may easily be examined within percolation lattices. Because each site has an occupation probability that is independent of the surrounding sites, the model can be made "periodic" simply by changing site
connectivity to be periodic at the lattice boundaries. Specifically, if a lattice of size $L \times L$ sites has coordinates $(i, j)$, and if both sites $(0, b)$ and $(L, b)$ are occupied, they are defined to be part of the same cluster for all indices $b$. Similarly, sites $(c, 0)$ and $(c, L)$ are also part of the same cluster when both are occupied for all $c$.

In this case, as Fig. 8 shows, size distributions in periodic percolation lattices remain strongly influenced by the finite lattice size, appearing qualitatively similar to those measured in non-periodic lattices with truncated clusters included in the size
distribution (Fig. 5). That is, distributions have a local maximum for cluster areas that are similar to the area of the domain. Such a local maximum is an example of a non-power law size distribution that is not representative of the power law cluster size distribution that is known to characterize a larger lattice. The implication is that periodic boundary conditions cannot be adopted as a fix for finite domain effects on a measured size distribution.

### 3.4   Distributions other than a power law

Even if the distribution of object sizes does not follow a power law, domain truncation effects may still bias measured size distributions. As an example, consider the distribution of raindrop sizes as measured by the new Differential Imaging Emissivity Distrometer (DEID). The DEID measures raindrop mass by measuring the time it takes for rain drops to evaporated after



landing on a hotplate (Rees et al., 2021). Water drop areas and lifetimes can be estimated from images of the hotplate, from which precipitation rates and size distributions can be estimated based on first-principles heat transfer physics. Because the

procedure requires calculating size distributions of droplets within a finite 2-D image, drop size distribution estimates may be affected by droplets truncated by the edge of the image in a similar manner to images of cloud fields taken by a satellite.

The main difference between precipitation and cloud size distributions is that precipitation size distributions tend to follow an exponential rather than a power law (Marshall, 1948; Singh et al., 2023). Nonetheless, removal of truncated droplets from the analysis would still influence the measured distributions. This can be illustrated by examining a manufactured exponential

distribution. For this purpose, we create a percolation lattice with site occupation probability just smaller the critical probability $p_c$. In this case, analytical results suggest that cluster sizes follow a power law with an exponential tail (Stauffer, 1992). The characteristic cluster size of the exponential tail increases without bound as the site occupation probability approaches $p_c$.

Figure 9 shows histograms of cluster sizes calculated from percolation lattices with site occupation probability equal to 0.5 for several sizes of lattice subdomain. As shown in Appendix C, in this case the cluster size distribution is exponential for

clusters larger than $\sim 200$ sites. The fraction of truncated clusters, relative to the total for each bin, never exceeds 50% in the $10000 \times 10000$ and $250 \times 250$ lattices, indicating truncation effects are insignificant. However, a histogram taken from the $40 \times 40$ lattices is strongly influenced by the removal of truncated clusters, undersampling large clusters relative to sampling done within a larger domain.

As with power laws, undersampled bins in an exponential distribution are dominated by truncated clusters. Applying the

same 50% truncated cluster criterion provides a straightforward method to identify which bins are most influenced by the choices of including or removing truncated clusters. A more accurate size distribution can still be obtained provided that these bins are omitted from the fit.

## 4    Conclusions

There is a significant disagreement in the literature on what should be the appropriate choice of distribution used to describe

cloud horizontal areas. Most studies find that cloud areas follow a power law $n(a) \propto a^{-(\alpha+1)}$, although there is considerable disagreement about the precise range of scales over which the power law applies, as well as what the precise value of $\alpha$ is. While some of this disagreement may be due to differences in local climatological characteristics, a recent study proposed some of the disagreement owes to the use of inferior linear regression-based fitting methods and argued Maximum Likelihood-based methods are superior (Savre and Craig, 2023).

The present study shows that the choice of fitting method cannot explain the disagreement among observations. We argue instead that Maximum Likelihood Estimation is inappropriate for cloud studies because its requirement of statistically independent data points is not satisfied. Instead, we find that a linear regression to logarithmically-spaced bins serves as an accurate fitting method to power law distributions, even those that do contain statistically independent data points, provided the simple requirement is adopted that bins with fewer than $\sim 24$ counts are omitted from the regression. Linear regression also



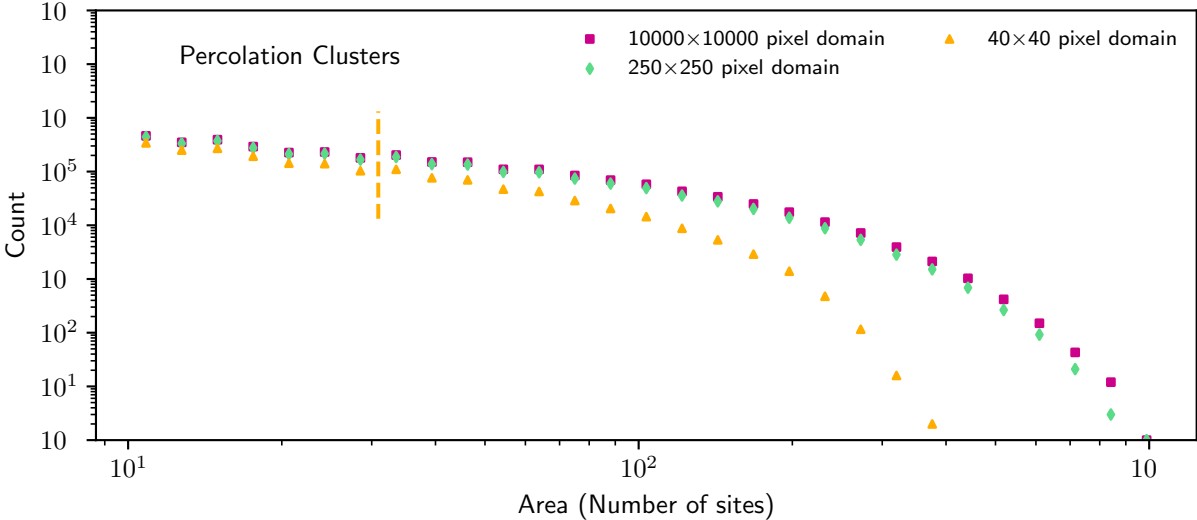

**Figure 9.** Histogram of percolation cluster areas generated in lattices with site occupation probability equal to 0.5. Plotted counts do not include truncated clusters. For the $40 \times 40$ lattices, as in Fig. 5, truncated clusters outnumber non-truncated clusters in bins to the right of the dashed yellow line. For the larger domains, there are no bins that contain a majority of truncated clusters.

has the advantage of being computationally trivial and more conceptually straightforward than Maximum Likelihood-based alternatives.

We propose that different accounts of cloud power law behavior in the literature are best explained by how clouds are treated that span the edge of the measurement domain and are "truncated". The removal of truncated clouds from the distribution introduces an artificial "cutoff scale" beyond which clouds can be significantly undersampled. If on the other hand, truncated clouds are included, a local maximum in the distribution appears at areas comparable to the domain scale that is not reflective of the true distribution. Even in periodic simulations, measured size distributions are not representative of size distributions obtained in larger domains. If truncation effects are not considered, a power law may still be measured but the value of the power law exponent may be underestimated by as much as 36% or overestimated by as much as 24%.

Fortunately, truncation effects are most important for only the largest clouds in a size distribution. The particular affected scale is easily identified by counting, for each bin, the fraction of clouds that are truncated relative to the total. We recommend that power law fits be applied only to bins in which the fraction of truncated clouds is less than 50% of the total for accurate measurements of the power law exponent.

Truncation effects are not limited to power law size distributions, and we show how exponentially-distributed objects can be similarly affected. Fortunately, the 50% truncated object criterion works regardless of the underlying form of the distribution.

The issues and remedies discussed here are not specific only to atmospheric clouds. They could just as easily be applied to size distributions characterizing any other phenomena if they are measured within a finite geometric domain, for example



ecological predator-prey models (Pascual et al., 2002), CO$_2$ pockets in sedimentary rocks (Iglauer et al., 2010), snowflakes (Rees et al., 2021), cloud droplets (Beals et al., 2015), aerosols (Magín Lapuerta and Gómez, 2003), and soil particles (Mora et al., 1998).

## Appendix A:  Statistical variability in histogram bin counts

The result that linear regression-based fitting methods can accurately estimate the power law exponent $\alpha$, provided that bins with counts less than $\sim 24$ are omitted from the regression, might appear to contradict the results of Clauset et al. (2009). They argued that linear regression-based estimation methods for $\alpha$ are biased in their Appendix A. In this section, we explain their argument, why linear regression can in fact be accurate, and point out a subtle error made in the widely cited work by Clauset et al. (2009).

The central issue is the statistical error of bin counts in a histogram. As a conceptual model, consider a large number of experiments that each measure some variable many times and bin results into a histogram. The count in each bin can be expected to be roughly similar experiment-to-experiment but not exactly the same. The "statistical error" is the standard deviation of the bin counts, which could be estimated for example by sampling a large collection of experiments.

This conceptual model can be made more precise by considering the experiments as a random counting process consisting of $N$ independent and identically distributed draws of a random variable $X$ from an arbitrary distribution. Consider some bin $i$ of fixed size and location in the parameter space of $X$. The "bin count function" $n_i$ may be introduced by first considering an indicator function $\mathbb{I}$ which is equal to 1 if $X$ lies within $i$ and 0 otherwise. The bin count $n_i$ is simply the sum of $\mathbb{I}$ over all draws.

The advantage of introducing $\mathbb{I}$ is that the Central Limit Theorem applies to $\mathbb{I}$ even if it does not to $X$. Specifically, the theorem requires independent, identically distributed random variables, finite variance, and finite mean. Because the mean value $\langle \mathbb{I} \rangle$ and the variance $\sigma^2$ of $\mathbb{I}$ are both bounded by 0 and 1, these assumptions are satisfied and therefore the Central Limit Theorem states that the bin count $n_i$ tends to a Gaussian distribution as $N \to \infty$.

Standard linear regression packages assume each data point has Gaussian error. In their Appendix A, Clauset et al. (2009) argued that linear regression-based estimation methods are invalid if the regression is performed to $\log n_i$, which is supposedly *not* Gaussian if $n_i$ is Gaussian.

This is incorrect because the Central Limit Theorem also states that the variance of $n_i$ tends to $\sigma^2 \sqrt{n_i}$ (where $\sigma^2 \leq 1$ because $0 \leq \mathbb{I} \leq 1$), and so the standard deviation of $n_i$ is $\sigma n_i^{1/4} \ll n_i$ for large $n_i$. This means almost all errors $\varepsilon$ are much smaller than $n_i$, and so we may linearize $\log\left(n_i + \varepsilon\right)$ using a Taylor expansion about $n_i$ so

$$\log\left(n_i + \varepsilon\right) \propto \log n_i + \frac{\varepsilon}{n_i} + \mathcal{O}\left(\frac{1}{n_i^2}\right). \tag{A1}$$

Thus, in the neighborhood of $n_i$, the logarithmic transformation is linear if terms of order $1/n_i^2$ are neglected. Because the transformation is linear, $\log n_i$ is also Gaussian distributed, in which case linear regression packages estimate both errors and



the power law exponent itself accurately for large $n_i$. The requirement of large $n_i$ is not a significant requirement because it is in fact required even for $n_i$ to be Gaussian, because $n_i$ is a discrete quantity.

Figure 3 shows an empirical test of the above reasoning, where 1000 samples, each containing 5000 randomly generated numbers, were drawn from a power law distribution with $\alpha = 1$. The bin $10 < x_i < 100$ showed Gaussian variability in the bin count $n$ as well as the log of the bin count $\log n$ as illustrated by nearly identical Kolmogorov-Smirnov p-values (0.333 vs. 0.326, respectively). Table A1 shows Kolmogorov-Smirnov p-values for more combinations of bin locations and sample size.

    We suggest this result explains why the linear regression technique used in Section 2 is accurate. Previous results, includ-
ing those by Clauset et al. (2009), may have produced biased power law exponents simply because they included bins with small $n_i$ in the linear regression. If such bins are excluded from the linear regression, statistical errors of $\log n_i$ are approximately Gaussian distributed and estimations of power law exponents can be accurately estimated within normal measurement uncertainties.

    The above argument applies for measurements that are statistically independent. As argued in Sect. 2, cloud measurements
cannot be statistically independent due to global limits on moisture, energy, and space. In this case, statistical errors of $n_i$ are not Gaussian. A priori, one might expect statistical errors to be log-normal instead (implying $\log n_i$ is Gaussian), because scale-by-scale conservation of a relevant variable $\phi$ implies $\phi n_i$ is constant (because the total amount $\Phi$ within a bin is $\phi n_i$). As an example, Garrett et al. (2018) identified cloud perimeters $p$ as controlling cloud formation in thin quasi-horizontal layers. By assuming $p n_i = const.$, they derived a power law distribution for cloud perimeters. Similarly, in their Sect. 3.3 Lovejoy and
Schertzer (2018) argue for a "multiplicative central limit theorem", which implies the logarithm of the energy flux is Gaussian distributed.

    Regardless, lognormality in statistical errors of $n_i$ is a convenient assumption when using linear regression-based methods for estimating a power law exponent, because in this case $\log n_i$ is Gaussian as software packages assume. However, it is conceivable that statistical errors of cloud area measurements might follow a different distribution, in which case neither
Maximum Likelihood Estimation nor linear regression would be strictly appropriate.





**Table A1.** Kolmogorov-Smirnov p-values, as in Fig. 3, for more combinations of bin location, sample size, and $\alpha$. Combinations are excluded if the mean $n_i$ is less than 3. With two exceptions, every case where the null hypothesis of normality in $\log n_i$ would be rejected under a 95% confidence interval (bold) has a mean count that is less than 24. In the two exceptions, the null hypothesis would still be rejected if the confidence interval was raised to 99%. This is roughly consistent with the expectation that 1 in 20 experiments would result in a false conclusion using a 95% confidence interval.

| Bin location $i$ | Sample size | $\alpha$ | Mean $n_i$ | Linear p-value | Logarithmic p-value |
|---|---|---|---|---|---|
| (9, 10) | $10^3$ | 1 | 11 | $\mathbf{1.3 \times 10^{-6}}$ | $\mathbf{0.7 \times 10^{-11}}$ |
| (9, 10) | $10^4$ | 1 | 111 | 0.3 | $0.5 \times 10^{-1}$ |
| (9, 10) | $10^4$ | 2 | 24 | $\mathbf{1.2 \times 10^{-3}}$ | $\mathbf{0.6 \times 10^{-4}}$ |
| (9, 10) | $10^5$ | 1 | $1.1 \times 10^3$ | 0.9 | 0.7 |
| (9, 10) | $10^5$ | 2 | 234 | 0.5 | 0.3 |
| (9, 10) | $10^6$ | 1 | $1.1 \times 10^4$ | 0.6 | 0.5 |
| (9, 10) | $10^6$ | 2 | $2.3 \times 10^3$ | 0.7 | 0.5 |
| (10, 100) | $10^3$ | 1 | 90 | $1.2 \times 10^{-1}$ | $0.6 \times 10^{-1}$ |
| (10, 100) | $10^3$ | 2 | 10 | $\mathbf{0.3 \times 10^{-6}}$ | $\mathbf{0.2 \times 10^{-9}}$ |
| (10, 100) | $10^4$ | 1 | 901 | 0.9 | 0.9 |
| (10, 100) | $10^4$ | 2 | 99 | 0.4 | $\mathbf{0.2 \times 10^{-1}}$ |
| (10, 100) | $10^5$ | 1 | $9.0 \times 10^3$ | 0.9 | 1.0 |
| (10, 100) | $10^5$ | 2 | 991 | 0.6 | 0.8 |
| (10, 100) | $10^6$ | 1 | $9.0 \times 10^4$ | 0.9 | 0.9 |
| (10, 100) | $10^6$ | 2 | $1.0 \times 10^4$ | 0.9 | 0.9 |
| (99, 100) | $10^5$ | 1 | 10 | $\mathbf{0.3 \times 10^{-5}}$ | $\mathbf{1.1 \times 10^{-10}}$ |
| (99, 100) | $10^6$ | 1 | 101 | $1.0 \times 10^{-1}$ | $1.1 \times 10^{-1}$ |
| $(10^2, 10^3)$ | $10^3$ | 1 | 9 | $\mathbf{0.3 \times 10^{-5}}$ | $\mathbf{0.8 \times 10^{-13}}$ |
| $(10^2, 10^3)$ | $10^4$ | 1 | 90 | $1.4 \times 10^{-1}$ | $0.6 \times 10^{-1}$ |
| $(10^2, 10^3)$ | $10^5$ | 1 | 901 | 0.8 | 0.6 |
| $(10^2, 10^3)$ | $10^5$ | 2 | 10 | $\mathbf{0.2 \times 10^{-6}}$ | $\mathbf{0.2 \times 10^{-10}}$ |
| $(10^2, 10^3)$ | $10^6$ | 1 | $9.0 \times 10^3$ | 0.6 | 0.5 |
| $(10^2, 10^3)$ | $10^6$ | 2 | 99 | $0.8 \times 10^{-1}$ | $1.2 \times 10^{-1}$ |
| $(10^3, 10^4)$ | $10^4$ | 1 | 9 | $\mathbf{0.5 \times 10^{-7}}$ | $\mathbf{1.4 \times 10^{-14}}$ |
| $(10^3, 10^4)$ | $10^5$ | 1 | 90 | 0.4 | $\mathbf{0.5 \times 10^{-1}}$ |
| $(10^3, 10^4)$ | $10^6$ | 1 | 899 | 0.8 | 0.9 |



## Appendix B: Correction algorithms for domain truncation effects

The method we propose to address domain truncation effects, namely to omit bins in which the truncated clouds are greater than 50% of the total, effectively removes the large portion of the size distribution. If the large portion is of interest, an algorithm could be derived in principle for the effects of the removal of clouds that are truncated by the domain edge.

Consider, for example, the case of cloud area distributions. If cloud locations are statistically independent of the domain edge location, the probability a cloud is truncated by the domain edge $P_{\text{truncated}}(a)$, a function of cloud area $a$, can be calculated from the mean cloud "lengths", defined as the longest distance from one end of the cloud to the other in the orthogonal $x$ and $y$ dimensions of an image. If cloud lengths can be related to measured cloud areas, a correction for the removal of clouds touching the edge is straightforward to implement since $n(a)_{\text{obs.}} = (1 - P_{\text{truncated}}(a))\, n(a)$ where $n(a)$ is the true cloud area

distribution. Wood and Field (2011) used a similar formulation, assuming clouds were square-shaped in order to relate cloud areas to cloud lengths.

In general, obtaining an appropriate correction algorithm can be a surprisingly difficult problem. For clouds specifically, there are several issues. First, cloud lengths would likely not be proportional to $\sqrt{a}$ since clouds are fractal and the length dimensions of fractal objects do not necessarily scale with $\sqrt{a}$ (Mandelbrot, 1982). Secondly, on a rotating planet cloud lengths

in the zonal direction may be related to area through a different function than cloud lengths in the meridional direction, since there are different temperature, moisture, and Coriolis force gradients zonally vs. meridionally, and these gradients be functions of horizontal scale. Thirdly, cloud locations are not statistically independent of domain edge location for large domains due to variability in regional climatological cloud fraction owing to e.g. the placement of the continents or the sphericity of the Earth. Finally, cloud shapes are quite variable, and so any relationship between cloud length and cloud area can only be expressed

statistically.

This last point is particularly problematic, since it makes simply measuring the relationship between cloud area and cloud length difficult and affected, again, by choice of the domain size. Consider a hypothetical case where most large clouds are much longer zonally than they are meridionally, but that areas are measured in a square domain. The only clouds whose zonal lengths can be accurately estimated are those not truncated by the West or East sides of the domain. Such clouds will

be predominately *not* wider zonally than meridionally because the zonally wider clouds will be truncated and subsequently removed from the analysis. The measured sample will be heavily biased away from zonally wide clouds, skewing the measured relationship between cloud length and area.

For a more in depth exploration of the subtleties involved with correcting object size distributions, see chapter 4 of the M.S. thesis by DeWitt (2023).

## Appendix C: Validation of exponential distributions of percolation clusters

To create an exponential distribution of cluster sizes, in Sect. 3.4 we create percolation lattices with site occupation probability equal to 1/2. Theoretically, this should result in a cluster size distribution that follows a power law with an exponential cutoff.





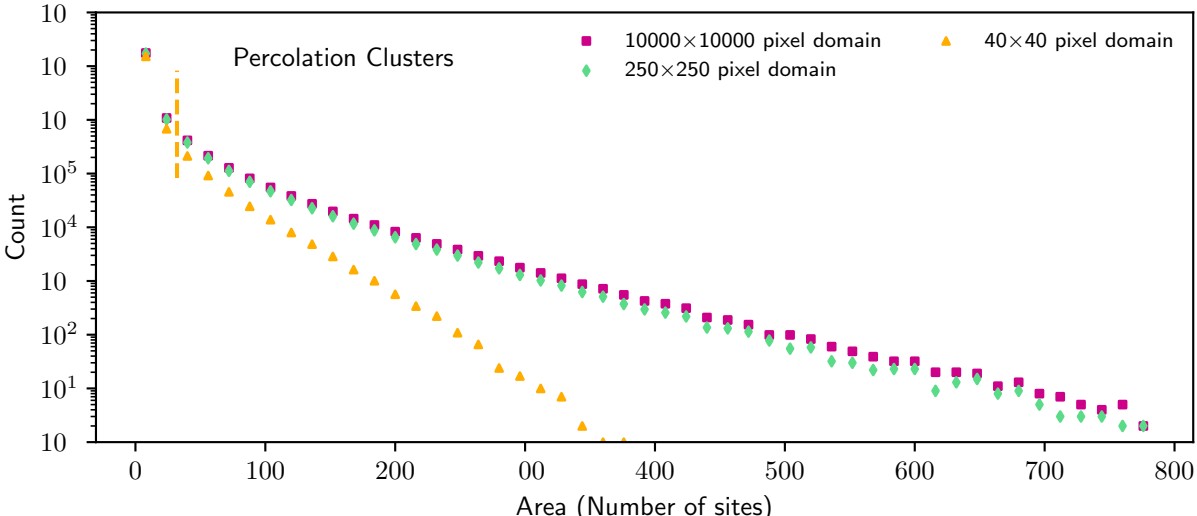

**Figure C1.** As in Fig. 9, but with the $x$-axis on a linear scale.

This is supported by Fig. C1, which shows the calculated histograms are indeed linear on a log-linear plot for $a \gtrsim 200$ sites, which is a requirement of an exponential distribution.

Interestingly, the $40 \times 40$ subdomains, which are strongly influenced by the removal of truncated clusters, also result in an apparently exponential distribution but with a steeper slope. Such exponential behavior could not continue to arbitrarily large cluster areas, however, because a pure exponential tail would predict nonzero probability of observing a cluster that is larger than the lattice iself, which is impossible if truncated clusters are removed.





## Appendix D: Tables of linear regression failure rates

Tables D1 and D2 displays failure rates for select linear regression-based estimators for $\alpha$ that are plotted in Fig. 2.



**Table D1.** Rate of reliable estimates of the power law exponent $\hat{\alpha}$ for different linear-regression-based estimation methods ("estimators") for 200 samples. Table D2 shows additional estimators for minimum bin counts of 30 and 50. Dashes indicate estimators where at least one sample did not contain bins spanning the required 1 order of magnitude after the minimum bin count threshold was applied, and thus the power law exponent could not be estimated. Errors $\varepsilon$ are estimated as two standard errors on the regression and coorespond to a 95% confidence interval. Biased estimators, defined as estimators whose failure rate is more than 5%, are marked in bold.

| Minimum bin count | Sample size | Number bins | Failure rate | Mean $\hat{\alpha}$ | Mean $\varepsilon$ |
|---|---|---|---|---|---|
| 0 | 1000 | 30 | 0.5 | 1.0 | 0.2 |
| 0 | 1000 | 100 | **10.5** | 0.9 | 0.2 |
| 0 | 1000 | 300 | **100.0** | 0.7 | 0.1 |
| 0 | 3000 | 30 | 0.5 | 1.0 | 0.1 |
| 0 | 3000 | 100 | 0.0 | 1.0 | 0.1 |
| 0 | 3000 | 300 | **78.0** | 0.9 | 0.1 |
| 0 | 10000 | 30 | 0.0 | 1.0 | 0.1 |
| 0 | 10000 | 100 | 0.5 | 1.0 | 0.1 |
| 0 | 10000 | 300 | 0.0 | 1.0 | 0.1 |
| 10 | 1000 | 30 | 2.0 | 0.9 | 0.2 |
| 10 | 1000 | 100 | 3.5 | 0.7 | 0.3 |
| 10 | 1000 | 300 | - | - | - |
| 10 | 3000 | 30 | 0.0 | 1.0 | 0.1 |
| 10 | 3000 | 100 | **7.0** | 0.9 | 0.1 |
| 10 | 3000 | 300 | **13.5** | 0.8 | 0.2 |
| 10 | 10000 | 30 | 1.0 | 1.0 | 0.1 |
| 10 | 10000 | 100 | 5.0 | 1.0 | 0.1 |
| 10 | 10000 | 300 | **37.0** | 0.9 | 0.1 |



**Table D2.** Continuation of Table D1 for minimum bin counts of 30 and 50.

| Minimum bin count | Sample size | Number bins | Failure rate | Mean $\hat{\alpha}$ | Mean $\varepsilon$ |
| --- | --- | --- | --- | --- | --- |
| 30 | 1000 | 30 | - | - | - |
| 30 | 1000 | 100 | - | - | - |
| 30 | 1000 | 300 | - | - | - |
| 30 | 3000 | 30 | 0.0 | 1.0 | 0.1 |
| 30 | 3000 | 100 | - | - | - |
| 30 | 3000 | 300 | - | - | - |
| 30 | 10000 | 30 | 0.0 | 1.0 | 0.1 |
| 30 | 10000 | 100 | 0.0 | 1.0 | 0.1 |
| 30 | 10000 | 300 | - | - | - |
| 50 | 1000 | 30 | - | - | - |
| 50 | 1000 | 100 | - | - | - |
| 50 | 1000 | 300 | - | - | - |
| 50 | 3000 | 30 | 0.0 | 0.9 | 0.1 |
| 50 | 3000 | 100 | - | - | - |
| 50 | 3000 | 300 | - | - | - |
| 50 | 10000 | 30 | 1.5 | 1.0 | 0.1 |
| 50 | 10000 | 100 | 0.5 | 1.0 | 0.1 |
| 50 | 10000 | 300 | - | - | - |





*Author contributions.* TDD: conceptualization, formal analysis, methodology, writing (original draft preparation). TJG: conceptualization, funding acquisition, supervision, methodology, writing (review and editing).

*Competing interests.* At least one of the (co-)authors is a member of the editorial board of Atmospheric Chemistry and Physics.

*Code and data availability.* Software code used to generate, analyze, tabulate, and plot all data described in the text is available at
https://github.com/thomasdewitt/Size-distributions-in-finite-domains. The GOES-West dataset was downloaded from the ICARE Data Center in Lille, France (ICARE).

*Acknowledgements.* Karlie N. Rees, Steven K. Krueger, and Corey Bois all contributed to discussions about the research. The Center for High Performance Computing at the University of Utah provided data storage and computing services. This research has been supported by the National Science Foundation Grant No. PDM-2210179.



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
