# Peer review of "Finite domains cause bias in measured and modeled distributions of cloud sizes"

_EGUsphere, 2024_

## Author Comment (AC1)

We thank both George Craig and Theresa Mieslinger for their helpful comments.

In this document, italics denote reviewer comments and the grey boxes contain excerpts of the manuscript where changes were made. Blue indicates added text and  in keeping with the changes file.

**Response to George Craig's comments (Reviewer 1)**

*"1. Appendix A shows that the argument of Clauset et al. that applying linear regression in log-log space was inappropriate because of the implicit assumption about the error distribution does not apply if there is a sufficient amount of data. The central limit theorem implies that the error distributions will converge to Gaussian in either space. So there is no reason to expect MLE to give better results than LR. The paper also argues that the assumptions underlying MLE are violated for cloud fields (l.116ff). This seems reasonable but also holds for LR; in particular the heteroscedasticity associated with finite domain size described here is also a violation of the assumptions for least square fitting. As far as I can see, the only argument that actually prefers LR to MLE is in Appendix A (l.385). This is the suggestion by Lovejoy et al. that perhaps errors in cloud sizes are lognormally distributed and therefore normal in log space. But the Appendix goes on to conclude, correctly, that we don't really know what the error distributions are, so without examples of one method producing better results that the other, it seems inappropriate to draw conclusions about one method being more appropriate than another."*

The point about the heteroscedasticity in particular is good and one we had not considered. Because the effect of the domain size on the bin count increases with cloud size, bin count errors may be larger for larger cloud sizes. This could be problematic for either LR or MLE. Presumably, removing the largest bins as recommended for domain size effects would also help with problems with heteroscedasticity; however, it is unclear whether heteroscedasticity could still be problematic. We added discussion of this point in Appendix A, shown below.

As for preferring LR to MLE, it is conceivable that LR is more appropriate because it would apply for either normal or lognormal errors, whereas MLE would only be suitable for normal errors. The appendix section mentioned in the comment also hypothesizes that variability in cloud sizes could be lognormal. We believe the best argument in favor of LR is not statistical, but practical: LR is familiar and straightforward to implement. MLE, for a bounded power law, is comparatively computationally expensive and difficult in implement. This is because the likelihood function must be numerically maximized. However, we know of no rigorous arguments that cloud sizes have lognormal errors, and agree that the conclusions were stated too strongly. Overall, the main point is intendended not to be that LR is statistically superior to MLE, but that the effect of a finite domain is more important.

We made the following changes. First, in several places we removed claims of the superiority of LR:

Abstract:

> Here, we counter that  linear regression is both simpler and equally accurate, provided the simple precaution is followed that bins containing less than $\sim 24$ counts  are omitted from the regression.

Line 64:

> Here  we argue that the  choice of fitting method is less important than whether past studies properly accounted for the finite size of the study domain. A finite domain size is a general problem for measuring scale-free quantities. For example, Serafino et al. (2021) argued that scaling properties of networks can be obscured by such finite-size effects, causing a truly scale-free network to appear non-scaling.

Line 320:

> The present study shows that the choice of fitting method cannot explain the disagreement among observations.

The paragraph at line 116, mentioned in the above comment, was modified and moved to appendix A which discusses the issue of preferring LR more thoroughly. Appendix A was divided into two sections, the second aimed at this question and shown below. A discussion of heteroscedacticity was also added to this section.

**A.1 Are cloud sizes statistically independent?**

The above argument applies for measurements that are statistically independent because statistical independence implies the bin count $n_i$ has Gaussian error. The Maximum Likelihood Estimation method presented by Clauset et al. (2009) also requires statistically independent errors. Unfortunately, statistical independence is often not satisfied in physical systems such as naturally occurring networks (Serafino et al., 2021) and clouds (Garrett et al., 2018). Because cloud formation is constrained by the total available moisture, energy, and space, individual cloud areas are not physically independent and this appears in the statistics. For example, a large but rare cloud that covers over half of a given measurement domain makes it impossible to observe another similarly-sized cloud because a second large cloud could not fit inside of the domain. Thus, the first observation (i.e. the large cloud) alters the probability of the next observation, which violates statistical independence. Similarly, a finite amount of total available energy or moisture makes future cloud formation contingent on what has occurred in the past. Thus, statistical errors of $n_i$ may not be Gaussian, in which case Maximum Likelihood Estimation-based methods would be inappropriate.

A priori, one might expect statistical errors for cloud sizes to be log-normal instead (implying $\log n_i$ is Gaussian), because scale-by-scale conservation of a relevant variable $\phi$ implies $\phi n_i$ is constant (because the total amount $\Phi$ within a bin is $\phi n_i$). As an example, Garrett et al. (2018) identified cloud perimeters $p$ as controlling cloud formation in thin quasi-horizontal layers. By assuming $p n_i = const.$, they derived a power law distribution for cloud perimeters. Similarly, in their Sect. 3.3 Lovejoy and Schertzer (2018) argue for a "multiplicative central limit theorem" for energy flux, which implies the logarithm of the energy flux is Gaussian distributed.

Regardless, lognormality in statistical errors of $n_i$ is a convenient assumption when using linear regression-based methods for estimating a power law exponent, because in this case $\log n_i$ is Gaussian as software packages assume. However, it is conceivable that statistical errors of cloud area measurements might follow a different distribution, in which case neither Maximum Likelihood Estimation nor linear regression would be strictly appropriate. Another problem with either method could be heteroscedasticity: that is, the variance of $n_i$ could depend on cloud size. This could be due to either physical differences in spatial scale or the effect of the finite domain size. Further work is needed to determine both the shape of the distribution of $n_i$ as well as its dependence on spatial scale.

> *"2. The paper cites the work Savre and Craig (2023) (hereafter SC) as a motivation, but focuses exclusively on the use of MLE, ignoring other key aspects of the recommended methodology, namely the use of a goodness of fit test to identify the appropriate region to fit. In practice, I suspect that removing bins that do not contain a minimum number of points from the fit, as recommended here, and restricting the range of fitting using a goodness of fit test, as recommended by SC, will confine both methods to the regime where there is "enough" data, and the LR and MLE methods will give the same exponents. This seems to be the case for the examples presented here. "*

The main discussion of restricting the fitting region to where the bin counts are sufficiently large is in Sect. 2. For LR and MLE to both produce similar estimates, it is required that the fitting range for LR be

restricted to those bins with sufficient counts. For the data in that section, MLE would already accurately estimate the power-law exponent even if the fitting range was not restricted. This was shown by Clauset et al. (2009), although in that case the distribution was an unbounded power law. The result would hold, however, for a bounded power law, though the MLE estimator would be different.

As far as LR and MLE giving the same exponents, our original claim that LR is superior to MLE was inappropriate because we do not know what the distribution of errors for cloud sizes is, as pointed out by your other comments. The modifications we made in response to those comments alters one major claim in the paper: instead of LR producing more accurate estimates, LR and MLE both produce similar estimates. This is supported by Mieslinger et al. (2019) who tested both methods as well as our Table 2, which shows that both methods produce similar estimates once finite domain effects are accounted for. We made one further change to emphasize this point:

> Other studies that estimated $\alpha$ over a range of scales that exclusively included large bin counts (e.g Benner and Curry, 1998; Cahalan and Joseph, 1989; Wood and Field, 2011; DeWitt et al., 2024) may have obtained estimates of $\alpha$ that were  as reliable as Maximum Likelihood-derived estimates.

The broader point made in the paper is that, even if the most statistically accurate methods were used including a goodness-of-fit test, truncation effects could result in an incorrect reported distribution. For the hypothetical scenario depicted in Fig. 7, a power-law with an exponential tail or a pure exponential would likely be the best fit with high confidence. The problem in this scenario would not be the statistical methods, but the interpretation of the reason for the exponential: it would be tempting to assume a physical limit on cloud sizes, whereas the truth is that the exponential is caused by a measurement artifact.

Thus, though they are an important tool, goodness-of-fit tests are not the subject of this paper. For the synthetic data in Sect. 2, MLE is already accurate without restricting the fitting range. In Sect. 3, goodness-of-fit tests would not be able to identify truncation effects. Truncation effects modify the data prior to any fit calculation.

> *"3. The paper apparently argues that fitting is not main cause of variation in estimated exponents, but that finite domain effects could be. It is shown that the use of MLE vs LR is not relevant, but there is still a potential sensitivity to the fitting methodology in that bins with insufficient data must be rejected to obtain robust results. Both issues could contribute to the diversity of exponents found in the literature, but it seems difficult to say much about their relative importance without reanalyzing previous data sets. It may not even be possible to separate the two effects cleanly, since as noted in text (l.257), bins with truncated clouds can coincide with bins that have few clouds, and both will be eliminated together. And of course different studies examine different meteorological situations and a diversity of distributions may be the correct answer. Given that some authors (e.g. Heus and Seifert 2013) have claimed to have checked for domain size effects, statements like "Finite domain effects are sufficient to account for previously observed discrepancies among reported cloud size distributions" (l.9) don't seem justified."*

Thank you for the comment. We do not intend to claim that finite domain effects are the only reason for differences, but that they could be a main contributor. We updated the sentence mentioned to reflect this point:

>  A much more significant and under-appreciated source of error  is how to treat clouds that are truncated by the edges of unavoidably finite measurement domains.

While fitting methodology could also contribute to differences in reported exponents, it is more difficult to see how fitting methodology could explain differences in reported scale break locations. This is discussed

in the text at line 168 (175 in the old version). However, the scale break introduced by removing truncated clouds bears strong resemblance to many previous reported distributions, and it is entirely possible that some of these scale breaks were caused by truncation effects. We added "particularly the range of scales over which a power law applies" to this statement in the conclusions to address this point. (The sentence was also modified in response to the first comment).

> The present study shows that the choice of fitting method cannot explain the disagreement among observations.  , particularly for the range of scales over which a power law applies. We

**Minor Comments**

> *"l.6 See major comment. Also the phrase "physical objects like clouds" is odd - like clouds in what respect? "*

This line was updated as discussed above. The phrase "physical objects like clouds" was removed.

> *"l.57 Would it be possible to come up with another example where assumptions of the fitting algorithm are not met. It was a bit confusing to have finite size effects introduced at this point in the paper. "*

Thank you for the comment, but unfortunately there may have been a typo because line 57 does not mention finite size effects or fitting algorithms. We are unsure of what location the comment refers to.

> *"l.60 SC do not simply argue that "the lack of consensus among prior measurements of cloud sizes owes to the use of inaccurate statistical methods to fit power law distributions." They also show that there can be real physical differences in the distributions, for example associated with the diurnal cycle. "*

This is a good point. We rephrased the text as follows:

> The lack of consensus among studies on the value of $\alpha$ may owe to differences in the dominant cloud type that was considered, or to how diurnal variability affects $a_{\max}$ (van Laar et al., 2019). But even if temporal and spatial variability of the size distribution exists, there remains a necessary prerequisite to measuring such variability, which is to first ensure the size distribution is being accurately measured in the first place. To this end, Savre and Craig (2023) recently argued that , while size distributions do show some variability, the use of inaccurate statistical methods to fit power law distributions could also partially explain the lack of consensus among prior measurements of cloud sizes. In particular, they showed that the common method of fitting a least-squares linear regression to a logarithmically-transformed histogram of cloud areas can lead to biased measurements of $\alpha$.

We also rephrased Line 317 in the conclusions to better represent Savre and Craig (2023)'s findings:

> . A recent study proposed that, while differences in local climatological characteristics  contribute to variability, some of the disagreement owes to the use of inferior linear regression-based fitting methods , arguing that Maximum Likelihood-based methods are superior (Savre and Craig, 2023).

> *"l.124 It seems unlikely that any fitting procedure on real cloud data can be proven to be statistically optimal - see major comment 1. "*

Thank you. This sentence has been moved and reworded as described above.

> *"Fig. 2 typo in x-axis label "30" "*

Thank you for pointing this out. This and similar typos in other figs (3,4,5,6) occured when the PDF was uploaded. We will make sure this issue is handled before publication.

> *"l.175ff The formulation of the problem in this paragraph seems to assume that there is a universal distribution of cloud sizes that would be seen in all the studies if it were not for methodological problems with fitting and domain size. One might argue that the hypothesis of a universal distribution has not yet been conclusively disproved by the diversity of observed distributions due to the potential methodological problems. "*

Thank you for pointing this out, because we do not necessarily want to claim that such a universal distribution exists. For example, some modeling studies have clearly demonstrated changes to the size distribution under different conditions (e.g. van Laar et al. (2019); Savre and Craig (2023)). Even if they did not completely account for the finite size of the domain, if the domain size is fixed while meteorological conditions vary, changes to the size distribution could not be explained by the domain size effect. We made the following change:

> Differences in the choice of fitting method used, whether Maximum Likelihood Estimation or regressions to linearly or logarithmically spaced bins, cannot explain these differences in measured $a_{\max}$.  While differences in meteorological conditions may contribute, meteorological influences may still be obscured by methodological problems. We next explore how the improper treatment of clouds that are truncated by the edge of the measurement domain could influence measured size distributions.

> *"l.230 "as being a real characteristic of clouds" change to "as also being a real characteristic of clouds under certain conditions" "*

Thank you, we made the suggested change.

> *"l.285 It's interesting that periodic BCs produce a peak in the size distribution near the domain size, similar to fits that include truncated clouds. Is there a reason for this? "*

The peak in the periodic percolation size distributions is certainly interesting. Intuitively, we think that the peak could be caused by a similar mechanism to the peak in the truncated case. Very large clusters "want" to form, but they cannot grow without intersecting themselves. However, as percolation clusters do not actually "grow", we feel this is a bit too squishy to mention in the text.

**Response to Theresa Mieslinger's comments (Reviewer 2)**

**Major comments**

> *"The truncation effect is quite obvious in Figures 4, 5, 6 and absolutely reasonable. However, cloud size distributions in previous literature rather show functional forms close to the ones present in this paper when truncated clouds are excluded. Could you please add a few words discussing this thought? "*

We agree that removing truncated clouds produces a distribution that bears a strong resemblance to many that have been previously reported in the literature. Given that few studies state whether truncated clouds are included or removed, we suspect many studies remove them without much consideration. We have added some citations in the text:

For each subdomain considered in the cloud imagery, if truncated clouds are removed from the size distributions, bin counts are increasingly undercounted at larger object areas as shown in Fig. **??**. A spurious scale break is introduced at these sizes that resembles an "exponential tail", a functional form suggested by Savre and Craig (2023) as being a real characteristic of clouds .—under certain circumstances. The form of the scale break also resembles many prior findings for both simulated and observed clouds (e.g. Cahalan and Joseph, 1989; Benner and Curry, 1998; Neggers et al., 2003; Heus and Seifert, 2013; Senf et al., 2018; van Laar et al., 2019; Christensen and Driver, 2021). . Locations of the spurious scale breaks, like those proposed in the literature, span several orders of magnitude but depend only on the domain size.

We also updated a portion of the conclusions in light of this comment:

We propose that different accounts of cloud power law behavior in the literature are best explained by  treatments of clouds whose geometries are "truncated" by the edge of the measurement domain. Removal of truncated clouds from the distribution introduces an artificial "cutoff scale" beyond which clouds can be significantly undersampled, with a resulting distribution consistent with many previous findings (e.g. Cahalan and Joseph, 1989; Benner and Curry, 1998; Neggers et al., 2003; Heus and Seifert, 2013; Senf230 et al., 2018; van Laar et al., 2019; Christensen and Driver, 2021). If included, a local maximum in the distribution appears at areas comparable to the domain scale that  does not reflect the true distribution.

*"For the 4000km x 4000km GOES domain considered, truncation effects seem small and the authors use this case as a reference, while truncation effects increase for smaller (sub-)domains. Also, the authors mention that not only domain size, but also the data resolution is an important factor (e.g. line 262-264). Is there a suggestion from the authors for a domain size - resolution combination or a minimal number of pixels needed to minimise truncation effects? This would make it easier to set previous literature into context where the authors argue that truncation is handled in a suboptimal way. "*

Thank you for the suggestion. Because the truncation effects occur at a scale controlled by the domain size, but the small end of the distribution is controlled by the pixel size, there is an effective minimum number of pixels required to accurately measure the power-law exponent. We have added the following paragraph:

Regardless of the domain size, truncation effects occur. For robust power-law fits, the resolution $\xi$ must be sufficiently small that the distribution spans the recommended two orders of magnitude (Stumpf and Porter, 2012) even after the 50% threshold is applied. For the square domains considered here, using a lower limit for the fit of $a_{\min} = 10\xi^2$, we find that the domain length $L$ must be of order $L/\xi \sim 300$ to satisfy this requirement.

We do not want to imply, however, that truncation effects only occur for small $L/\xi$. Any finite domain will show truncation effects, but the scale at which truncation effects occur changes with domain size. We updated the following paragraph to reflect this:

A scale break is introduced because larger clouds are more likely to be truncated and therefore to be removed from the analysis ( Fig. 5). This effect occurs for all domain sizes. The clouds need not be particularly large to be affected, as the scale break appears at  surprisingly small cloud areas occupying  between 1% and 0.1% of the subdomain area.

We also updated a sentence in the conclusion:

> While size distributions measured within any domain size are affected by truncation effects, they are most important  only for the largest clouds

> *"Only if less than 50% of all clouds in a given size bin touch the domain boundaries, the respective bin is taken into account for deriving a power law exponent. I was wondering how sensitive the results are to the 50% threshold and whether the authors tested lower/higher values. Could you add a sentence explaining this choice? "*

Thank you for the suggestion. We did indeed test other values, and added a sentence explaining why we settled on 50%.

> The 50% threshold represents a compromise between allowing for a significant range of scales to be analyzed but removing those bins most affected by truncation effects. A more stringent threshold of 10% (not shown) was found to produce similar results but omit a larger portion of the distribution from the fit.

> *"The under- and overestimations stated in the Conclusion in line 333 seem to be strongly related to the size of the subdomain which seem to be chosen rather arbitrarily. Could you explain why those are reasonable limits? Surely the over-/underestimation could be higher for even smaller domain sizes. "*

Thank you, this is correct. The size of the error certainly depends on the domain size and the fit range, and the particular case presented is merely meant to illustrate that large errors are possible and even when reported distributions appear to follow a power-law. We added some discussion of this point in the paragraph at line 245, which was also rephrased to address a minor comment:

> The simple remedy of calculating $\alpha$ by fitting a power law over a  relatively linear region of the distribution that is subjectively defined, as is often done, can lead to an overestimate of $\alpha$ if truncated clouds are removed  and an underestimate if they are included. As an example,  Fig. 7 depicts a hypothetical scenario  where the cloud area distribution is measured using images that each cover a domain $100 \times 100\,\text{km}$  in size. For this purpose, we use all $100 \times 100\,\text{km}$ subdomains from GOES. Values of $\alpha$  are calculated over a subjectively defined linear range of scales  for both cases of including and excluding truncated clouds in the distribution. Regardless of whether least-squares linear regression or Maximum Likelihood Estimation is used, including truncated clouds in the fit for $\alpha$ leads to an underestimate of 36% and 19%, respectively, while excluding them leads to an overestimate of 24% and 20%, respectively, relative to values calculated for the full $4000 \times 4000\,\text{km}$ domain  (Table 1). These errors would be greater if larger area values were included in the fit or if the domain were smaller. Nonetheless, it is clear from Fig. 7 that both approaches remain well approximated by a power law distribution, and so the truncation effect could easily be missed  if only one approach was presented. This would lead to reported power law behavior with a value of $\alpha$ that is a significant departure from the true value that would have been measured if the domain had been larger.

Additionally, the conclusions were rephrased to indicate the calculated errors are not generally applicable but merely from an example intended to show how bias could occur:

> In all cases, a power law may still  easily be measured, but the value of the power law exponent  could be underestimated or overestimated by  20 to 30% or more.

Minor comments:

*"Line 145-147: could you justify the relaxation from two orders of magnitude to only one? I suppose you'd have fewer samples to base your statistics on and going for only one oder of magnitude is a compromise? "*

The one order of magnitude threshold was chosen as an overly conservative threshold. For example, if a future study did only fit a power law to data spanning a single order of magnitude, the study could still be confident that our recommended methodology would result in an accurate estimate of the power-law exponent. Best practice, however, would be to use a larger span to fit real-world data, especially if power-law behavior itself is in question. We clarified these points as shown below:

> A fit to each sample is then performed only if the remaining bins span at least one order of magnitude in $a$. This requirement is necessary for any fitting method because power law distributions fundamentally describe systems spanning many scales (Newman, 2005), but is less stringent than the two orders of magnitude span recommended by Stumpf and Porter (2012). Because the fitting accuracy is increased for datasets spanning a larger range of values, the one order of magnitude requirement used here represents a conservative threshold for the purpose of evaluating fitting methods using a known power law distribution. If power law behavior itself is in question, a larger span is required.

*"Line 223-228: could you add the reason for simulating a 10000 x 10000 percolation lattice instead of resampling the GOES lattice, i.e. 2000x2000 pixels? Intuitively I would have assumed that you would want to simulate the same pixel number. Also, how do the q-values for the percolation lattice and also the grid cells stated in line 227 fit to the pixel numbers stated in Figure 5? It seems unnecessarily complicated to not go for the same pixel numbers in theory and observations, but maybe there is a good reason for it. "*

Thank you for raising this point, the $q$ values listed at line 227 were incorrect. At a late stage, Fig 5 was modified so that the subdomains in the GOES and percolation lattices each covered the same area in units of km$^2$ and pixels, respectively. However, we agree that it makes more sense to have each subdomain contain the same number of pixels, so we have updated the figures and text accordingly.

We have retained the 10000×10000 percolation lattice and created four sizes of subdomains. The new subdomains contain the same number of pixels as the GOES domain and subdomains. The 10000×10000 percolation lattice is useful because it illustrates that truncation effects can even occur in very large lattices such as the 2000×2000 lattice. In fact, truncation effects would occur in any finite lattice.

> We use values of $q \in \{20, 100, 200\}$ $q \in \{5, 50, 200, 500\}$ for the percolation lattices and values of $q \in \{10, 40, 100\}$ for GOES images. Thus the percolation sub-arrays have side lengths of 500, 100, or 2000, 200, 50, or 20 grid cells, and the GOES which match the dimensions of the original GOES array and its sub-arrays have side lengths of 200, 50, or 20 pixels.

*"Line 247: what is meant by "hypothetical scenario for GOES cloud areas"? Is it simply one subset of the image or do ALL subdomains go into the curves? Related to that, are the numbers stated in the following lines 250-251 only for this example or representative error estimates? The authors use them later in the conclusion and it reads as they are upper/lower bounds for over- and underestimations due to truncation effects. If it's indeed only one image it could also easily happen that you sample two very different cloud regimes as your 4000km x 4000km domain includes large clusters as part of the ITCZ as well as small trade cumulus clouds. Related, the x-tick labels to Figure 7 seem odd as there are counts for negative cloud areas. Also the Figure caption together with the paragraph discussing that Figure (line 247ff) leaves it unclear to me how the subset is designed and whether it is representative. Please clarify. "*

Thank you for raising this point, it was not clear in the text and especially in the figure caption which was misleading. What was intended in the figure caption was not a subset of subdomains but rather a subset of the histogram. All subdomains were used. The improved figure caption is below, and the edits to the paragraph at line 245 are above under the related general comment. We also discuss the conclusions under the general comment.

> Example of how a measurement $\hat{\alpha}$ of the power law exponent could be biased by whether or not truncated clouds are included in the analysis. The histograms shown are  created using all $100 \times 100$ km subdomains from GOES. The same histograms are shown in Fig. 4 but spanning a wider range of scales. This  particular range of scales is heavily influenced by the choice of including truncated clouds. Fits for $\alpha$ are shown in Table 1.

There appears to have been an issue during the file upload which caused some elements of the PDF to disappear. There are no negative areas and the tick labels should be 30, 100, and 300.

> *"The title of the subchapter 3.4 seems a bit broad and could be sharpened to set the reader's expectations. It seemed to me that you rather test your suggested truncation fix in an exponential distribution and show that it works there, too. "*

As suggested we changed the heading to "Finite domain effects for exponential distributions"

> *"In line 262-262 the authors state that errors could be further reduced. To what extent did you test other domain sizes / resolutions and could you add further info or include "(not shown)" such that it becomes clear that this statement is based on an analysis rather than gut feeling? "*

This line has been removed in response to another comment. Still, the claim is based on the results for the percolation lattices (table 2), where the larger lattices produce measured exponents closer to the known value. This was however not explicitly stated.

> *"Comment to Appendix B: domain truncation effects is a major focus of this paper. I would suggest to move the first part of Appendix B (maybe the content of lines 400-406) to the main part of the paper, but I leave it up to the authors to decide whether that is appropriate. Also, it would make the paper even stronger if the suggested correction for truncation were applied. But I can also accept if that goes beyond the scope of the present paper. "*

A correction function is certainly desireable, and we spent considerable time trying to formulate an acceptable corrective formula, but ultimately decided that any such formula would include implicit assumptions about clouds that we would not wish to make. The central issue is relating cloud area to length. Because clouds are fractal, this is not as simple as one would expect, and one would need to make assumptions about their shape. Since the upper end of the size distribution is near scales where Coriolis forces become important, and the cloud spans very different climate regimes, it seems inappropriate to assume large clouds would be the same shape as smaller clouds, just larger.

It could even be dangerous to implement a correction algorithm without considering these subtle and implict assumptions, so we chose to not emphasize the possibility. For example, the simplistic algorithm used in Wood and Field (2011) assumed clouds were square-shaped. Possibly, this could significantly modify the distribution at scales where cloud shapes may change, for example where the Coriolis force becomes important.

We made one small change to emphasize the difficulty:

> If cloud lengths can be related to  cloud areas – which is a nontrivial problem due to fractal cloud geometries – a correction for the removal of clouds touching the edge is straightforward to implement since $n(a)_{\text{obs.}} = (1 - P_{\text{truncated}}(a)) \, n(a)$ where $n(a)$ is the true cloud area distribution.

**Formal comments / typos:**

> *"Figure 2 seems to have a typo in the x-tick labels at the minimum bin count threshold 30 "*

Thank you. As mentioned some elements of the PDF file appear to have disappeared during upload, which is something we will ensure does not make it into the final version.

> *"Please add in the caption to Figure 3 some reference to the "left" and "right" plot for clarity "*

We have updated the caption to read:

> Statistical error in measured counts $n_i$ (left) or $\log n_i$ (right) within a bin bounded by 10 and 100 for a collection of 1000 samples, each containing 5000 randomly-generated power law distributed random variables $x_i$ with exponent $\alpha = 1$ (Eqn. 1).

> *"Figure 4, 5, 6, and 7 have several missing superscript numbers in the x- and y-tick labels. Please correct. "*

This is also due to the dropping of some PDF elements during upload.

> *"Fig 6 is mentioned before Fig 5 in the text. Please switch to make it easier to follow and jump back and forth. "*

We made this change.

> *"Caption Table1: is there a word missing in the last sentence? "...between the two domain sizes and [methods?] is expressed in units..." "*

We are not sure if there is a word missing, but the sentence is not easy to read. It has been rephrased:

> For comparison, fits to clouds measured in the full domain are included as "truth". "Differences" is the difference  in $\hat{\alpha}$ between the two domain sizes. The difference is expressed in units of standard errors as calculated  from the subdomains.

> *"Typo in line 260: "... for a series of of subdomains created..." "*

Thank you, it has been corrected.

> *"Typo in caption to Table D1 in second last sentence: "coorespond" instead of "correspond" "*

Thank you, it has been corrected.

**References**

Benner, T. C. and Curry, J. A.: Characteristics of small tropical cumulus clouds and their impact on the environment, Journal of Geophysical Research: Atmospheres, 103, 28 753–28 767, 1998.

Cahalan, R. F. and Joseph, J. H.: Fractal statistics of cloud fields, Monthly weather review, 117, 261–272, 1989.

Clauset, A., Shalizi, C. R., and Newman, M. E.: Power-law distributions in empirical data, SIAM review, 51, 661–703, 2009.

DeWitt, T. D., Garrett, T. J., Rees, K. N., Bois, C., Krueger, S. K., and Ferlay, N.: Climatologically invariant scale invariance seen in distributions of cloud horizontal sizes, Atmospheric Chemistry and Physics, 24, 109–122, https://doi.org/10.5194/acp-24-109-2024, 2024.

Garrett, T. J., Glenn, I. B., and Krueger, S. K.: Thermodynamic constraints on the size distributions of tropical clouds, Journal of Geophysical Research: Atmospheres, 123, 8832–8849, 2018.

Lovejoy, S. and Schertzer, D.: The weather and climate: emergent laws and multifractal cascades, Cambridge University Press, 2018.

Mieslinger, T., Horváth, A., Buehler, S. A., and Sakradzija, M.: The Dependence of Shallow Cumulus Macrophysical Properties on Large-Scale Meteorology as Observed in ASTER Imagery, Journal of Geophysical Research: Atmospheres, 124, 11 477–11 505, https://doi.org/https://doi.org/10.1029/2019JD030768, 2019.

Newman, M. E.: Power laws, Pareto distributions and Zipf's law, Contemporary physics, 46, 323–351, 2005.

Savre, J. and Craig, G.: Fitting Cumulus Cloud Size Distributions From Idealized Cloud Resolving Model Simulations, Journal of Advances in Modeling Earth Systems, 15, e2022MS003 360, https://doi.org/https://doi.org/10.1029/2022MS003360, 2023.

Serafino, M., Cimini, G., Maritan, A., Rinaldo, A., Suweis, S., Banavar, J. R., and Caldarelli, G.: True scale-free networks hidden by finite size effects, Proceedings of the National Academy of Sciences, 118, e2013825 118, https://doi.org/https://www.pnas.org/doi/abs/10.1073/pnas.2013825118, 2021.

Stumpf, M. P. H. and Porter, M. A.: Critical Truths About Power Laws, Science, 335, 665–666, https://doi.org/10.1126/science.1216142, 2012.

van Laar, T. W., Schemann, V., and Neggers, R. A. J.: Investigating the Diurnal Evolution of the Cloud Size Distribution of Continental Cumulus Convection Using Multiday LES, Journal of the Atmospheric Sciences, 76, 729 – 747, https://doi.org/https://doi.org/10.1175/JAS-D-18-0084.1, 2019.

Wood, R. and Field, P. R.: The distribution of cloud horizontal sizes, Journal of Climate, 24, 4800–4816, 2011.